# BMP pathway regulation of insulin signaling components promotes lipid storage in *Caenorhabditis elegans*

James F. Clark[1,2], Emma J. Ciccarelli[1,2], Peter Kayastha[1], Gehan Ranepura[1], Katerina K. Yamamoto[1,2], Muhammad S. Hasan[1], Uday Madaan[1,2], Alicia Meléndez[1,2], Cathy Savage-Dunn[1,2]*

1 Biology Department, Queens College, City University of New York (CUNY), New York City, New York, United States of America, 2 Ph.D. Program in Biology, The Graduate Center, City University of New York (CUNY), New York City, New York, United States of America

* cathy.savagedunn@qc.cuny.edu

**Data Availability Statement:** All relevant data are within the manuscript and its Supporting Information files.

## Abstract

A small number of peptide growth factor ligands are used repeatedly in development and homeostasis to drive programs of cell differentiation and function. Cells and tissues must integrate inputs from these diverse signals correctly, while failure to do so leads to pathology, reduced fitness, or death. Previous work using the nematode *C. elegans* identified an interaction between the bone morphogenetic protein (BMP) and insulin/IGF-1-like signaling (IIS) pathways in the regulation of lipid homeostasis. The molecular components required for this interaction, however, were not fully understood. Here we report that INS-4, one of 40 insulin-like peptides (ILPs), is regulated by BMP signaling to modulate fat accumulation. Furthermore, we find that the IIS transcription factor DAF-16/FoxO, but not SKN-1/Nrf, acts downstream of BMP signaling in lipid homeostasis. Interestingly, BMP activity alters sensitivity of these two transcription factors to IIS-promoted cytoplasmic retention in opposite ways. Finally, we probe the extent of BMP and IIS interactions by testing additional IIS functions including dauer formation, aging, and autophagy induction. Coupled with our previous work and that of other groups, we conclude that BMP and IIS pathways have at least three modes of interaction: independent, epistatic, and antagonistic. The molecular interactions we identify provide new insight into mechanisms of signaling crosstalk and potential therapeutic targets for IIS-related pathologies such as diabetes and metabolic syndrome.

## Author summary

Systemic homeostasis depends on the coordinated action of different cells and tissues throughout the body in response to internal and external cues. This coordination is achieved in part by cellular communication through the production, release, and response to molecular signals in a process known as cell signaling. These signals and their response pathways are highly conserved among animal species; therefore, we are able to use genetic tools in a simple animal model to identify broadly relevant mechanisms of function. Here,

**Funding:** This work was funded in part by R15GM112147 to CSD (National Institutes of Health/NIGMS; https://www.nigms.nih.gov/) and by R15GM102846 to AM (National Institutes of Health/NIGMS; https://www.nigms.nih.gov/). The funders had no role in study design, data collection and analysis, decision to publish, or preparation of the manuscript.

**Competing interests:** The authors have declared that no competing interests exist.

we demonstrate that two conserved signaling pathways, the bone morphogenetic protein (BMP) and insulin pathways, crosstalk to control lipid storage. We identify molecular components that mediate this regulation and determine their sites of action. Furthermore, we show that these two signaling pathways have multiple modes of interaction in a variety of developmental and physiological contexts. Due to the devastating human diseases caused by misregulation of these pathways, including cancer, cardiovascular disease, diabetes, and metabolic syndrome, understanding the mechanisms of crosstalk is critical.

## Introduction

Whole organism lipid homeostasis requires accurate integration of diverse cellular and environmental cues. We have previously shown that bone morphogenetic protein (BMP) signaling in the nematode *Caenorhabditis elegans* regulates lipid accumulation and that this function depends on the activity of the insulin receptor (InsR) DAF-2 [1]. This system thus provides a useful model to determine the mechanisms of signaling crosstalk in the context of an intact organism. Insulin/IGF-1 signaling (IIS) is a well characterized regulator of homeostatic processes. In mammals, IIS regulates glucose uptake and metabolic homeostasis [2]. Disruptions to insulin balance, such as insulin resistance, can lead to multiple metabolic disorders, including obesity and type II diabetes [3]. BMPs, on the other hand, are members of the Transforming Growth Factor beta (TGFβ) family of peptide ligands best known for roles in development, growth, and differentiation [4, 5]. Our research, however, has defined a role for BMP signaling in lipid homeostasis, using the powerful genetic tractability and imaging tools available in *C. elegans*. BMPs are similarly emerging as regulators of lipid homeostasis in vertebrates. For example, BMP8 can trigger a thermogenic response in mature brown adipose tissue [6]. BMP7 regulates expression of insulin signaling components in C3H10T1/2 cells, leading to fat browning [7]. Tissue culture studies have shown that BMP2 and BMP6 regulate insulin sensitivity in adipose cells [8]. BMPs may play a role in insulin regulation and age-related insulin resistance [9]. Finally, circulating BMP9 levels are correlated with metabolic syndrome and insulin resistance [10]. These studies suggest a conserved interaction between BMP and IIS pathways that motivated us to seek further mechanistic insight.

The BMP signaling pathway in *C. elegans* includes: the ligand, DBL-1 [11]; the receptors, DAF-4 and SMA-6 [12, 13]; founding members of the Smad family of signal transducers, SMA-2, SMA-3, and SMA-4 [14, 15]; and LON-2, a glypican that negatively regulates DBL-1 interactions with its receptor [16]. DBL-1, the *C. elegans* BMP2/4 homolog, plays a major role in body size regulation, male tail development, mesodermal patterning, and lipid accumulation [1, 11, 17]. The IIS pathway in *C. elegans* uses a single insulin receptor DAF-2/InsR [18, 19] in conjunction with 40 insulin-like peptides (ILPs) to regulate multiple homeostatic functions through the control of transcription factors, such as DAF-16/FoxO [20–22] and SKN-1/Nrf [23]. DAF-2/InsR has prolific effects on the development and homeostasis of the worm; disruptions to DAF-2/InsR lead to phenotypes in dauer formation, longevity, stress tolerance, innate immunity, germline maintenance, metabolism, and autophagy [24–30]. Autophagy genes were shown to be required for the remodelling that occurs during dauer development and also for the increased accumulation of lipids displayed in *daf-2/InsR* mutants [30–32].

In this study, we identify INS-4/ILP as a mediator of BMP-IIS crosstalk in lipid metabolism. We observe that Smad signaling in the hypodermis (epidermis) regulates expression of *ins-4*, and that *ins-4* mutants express a high-fat phenotype similar to that of *daf-2* mutants, indicating that INS-4/ILP is a negative regulator of fat accumulation. We analyzed two transcription

factors that act downstream of DAF-2/InsR: DAF-16/FoxO and SKN-1/Nrf. We find that loss of *dbl-1* alters the subcellular localization of DAF-16::GFP and SKN-1::GFP, and adjusts their sensitivity to DAF-2/InsR in opposite directions. We show that overexpression of DAF-16/FoxO, but not of SKN-1/Nrf, from a multicopy transgene suppresses the low-fat phenotype of *dbl-1/BMP* mutants suggesting that *dbl-1/BMP* regulation of fat metabolism is mediated through DAF-16/FoxO. Moreover, a *daf-16* mutation, but not a *skn-1* mutation, was able to suppress the high-fat phenotype of *ins-4* mutants. Because of this crosstalk, we asked whether BMP signaling modulates other functions of IIS. We show an antagonistic effect between BMP signaling and IIS in a *daf-2/InsR* mutant background in dauer formation, aging, and autophagy induction. Together with our previously results on lipid stores and body size [1], this work establishes three potential modes of interaction between DBL-1/BMP and DAF-2/InsR: independent, as in body size; epistatic, as in lipid regulation; and antagonistic, as seen in dauer formation, aging, and autophagy induction.

## Materials and methods

### Nematode strains and growth conditions

*C. elegans* were maintained on *E. coli* (DA837) at 15˚C, 20˚C, and 25˚C as specified. The wild-type strain used in this study was N2. The strains used in this study are as follows: LT121 *dbl-1 (wk70)*, CB678 *lon-2(e678)*, CS24 *sma-3(wk30)*, CB1370 *daf-2(e1370)*, RB2544 *ins-4(ok3534)*, HT1693 *unc-119(ed3);wwEx63*, DA2123 *adIs2122*, MAH14 *daf-2(e1370);adIs2122*, TJ356 *zIs356[daf-16::gfp]*, LD1 *ldIs7[skn-1::gfp]*, CS683 *daf-2(e1370)sma-3(wk30)*, CS681 *daf-2 (e1370);lon-2(e678)*. Crosses were used to obtain: CS663 *sma-3(wk30);ins-4(ok3534)*, CS633 *sma-3(wk30);wwEx63*, CS661 *daf-2(e1370);dbl-1(wk70);adIs2122*, CS634 *daf-2(e1370);lon-2 (e678);adIs2122*, CS626 *dbl-1(wk70);adIs2122*, CS627 *lon-2(e678);adIs2122*, CS685 *dbl-1 (wk70);zIs356*, and CS679 *dbl-1(wk70);ldIs7*.

### INS-4p::GFP expression

Animals were grown at 20˚C and imaged as day 1 adults. Images were taken on a Leica Microsystems Confocal microscope using a 20X objective. Camera settings were maintained identically between all samples and trials. Compound images were produced from compiling Z-stack images. n>10 per strain repeated in triplicate.

### Oil Red O staining

Protocol adapted from [33]. Animals were collected at the L4 stage in PCR tube caps and washed three times in PBS. Worms were then fixed for 1 hour in 60% isopropanol while rocking at room temperature. The isopropanol was removed and worms were stained overnight with 60% Oil Red O (ORO) solution while rocking at room temperature. ORO was removed and the worms were washed once with PBS w/ 0.01% Triton and left in PBS. Worms were mounted and imaged using an AxioCam MRc camera with AxioVision software. Images were taken using a 40X objective. ORO stock solution was made with 0.25g ORO in 50mL isopropanol. Intensity of the post-pharyngeal intestine was determined using ImageJ software. Pixel intensity was measured in the green color channel of the images. Three measurements using a 50px by 50px area were taken for each worm with background intensity subtracted for each individual picture. Statistical comparisons (two-way ANOVA, one-way ANOVA with post-hoc Tukey's multiple comparisons test, and unpaired t-test) were performed using GraphPad Prism 7 software. For each experiment, n>20 per strain repeated in triplicate.

## qRT-PCR

Worms were grown at 20˚C until a large number of eggs were observed on plates. Worms were then washed off using M9 buffer and the remaining eggs were allowed to hatch for 4 hours. Worms were then collected and placed on new plates and grown at 20˚C until late L2 stage, collected and mRNA was extracted using the RNeasy mini kit (Qiagen). Reverse transcriptase and quantitative real time PCR were performed as previously described using *act-1* as an internal control [34]. Three independent biological replicates were analyzed.

## DAF-16::GFP and SKN-1::GFP localization assays

RNAi induction and feeding were carried out as described in [35, 36]. For DAF-16::GFP, L4 animals were placed on EZ worm plates supplemented with 1mM IPTG and 50mg/ml carbenicillin overnight. Adults were moved to new RNAi plates and allowed to lay eggs for 4–6 hours to synchronize the progeny. Animals were then allowed to develop at 15˚C until the L3 larval stage followed by a shift to 20˚C to bypass the dauer induction of *daf-2* knockdown. L4 animals were assayed for GFP localization. L4440 was used as the empty vector control. Images were taken using a Zeiss ApoTome with AxioVision software and a 40X objective. Exposure times were kept consistent across strains within each experiment. GFP localization was categorized as low, medium, or high. n>20 per strain, repeated in duplicate. For SKN-1::GFP, L4 animals were placed on EZ worm plates supplemented with 1mM IPTG and 50mg/ml carbenicillin seeded with HT115 *E. coli*. Animals were allowed to lay eggs overnight. Adults and L1 larvae were removed the following day with M9. Remaining eggs were synchronized with a 4 to 6 hour hatch, with the L1s being transferred to new RNAi plates. Animals were then allowed to develop at 15˚C until the L3 larval stage followed by a shift to 25˚C to bypass the dauer induction of *daf-2* knockdown. Animals were left at 25˚C overnight and then assayed for GFP localization. L4440 was used as the empty vector control. Images were taken using a Zeiss ApoTome with AxioVision software and a 20X objective. Exposure times were kept consistent across strains within each experiment. GFP localization was categorized as low, medium, or high. n>40 per strain, repeated in duplicate. Statistical comparisons (chi-square test) were performed using GraphPad Prism 7 software.

## Dauer formation assay

Five to ten L4 animals were placed per plate to lay eggs overnight at 15˚C. Adults and L1s were removed on the subsequent day with M9 buffer. Plates with remaining eggs were then placed at either 20˚C or 25˚C to develop. Animals were then observed 48 hours later; dauer-like larvae were tallied versus L4s and adults. Statistical comparisons (unpaired t test and two-way ANOVA) were performed using GraphPad Prism 7 software. n>50 per strain at each temperature, repeated in triplicate.

## Lifespan assay

Lifespan plates were prepared at least one day prior to plating. Each plate was seeded with 400 μl of E. coli, and 50 μM 5-Fluoro-2'-deoxyuridine (FUdR) was added to inhibit progeny production that would otherwise cause bagging (death due to internal hatching of progeny) in some genotypes. On Day 0, >100 L4 worms of each genotype were picked, 20 worms to a plate. The worms were moved to new FUdR plates approximately once per week. The number of surviving and deceased worms were counted every other day. Deceased worms were identified as those that exhibited no movement in response to a gentle tap from the platinum wire pick. After counting, deceased worms were removed from the plates. Animals that were not found were censored. Three trials were performed.

### GFP::LGG-1 autophagy assay

Autophagy was assayed via the formation of GFP punctae as described in [37]. Animals were grown at 20˚C and imaged at the L3 larval stage; dauer larvae were not imaged or quantified. GFP::LGG-1 punctae were counted in the seam cells of the hypodermis. Images were taken using a Zeiss ApoTome with AxioVision software and a 100X objective. Exposure times were kept consistent across strains within each experiment. Statistical comparisons (unpaired t test and one-way ANOVA) were performed using GraphPad Prism 7 software. n = 15 per strain, repeated in duplicate.

### SQST-1::GFP autophagy assay

Impaired function of the ribosomal protein RPL-43 causes the intracellular accumulation of SQST-1 aggregates, particularly in the intestine, which can be removed upon autophagy induction [38]. *rpl-43(bp399)* mutant adults were placed on RNAi plates (including 1mM IPTG and 50mg/ml carbenicillin) and their progeny were allowed to develop until the L4 stage, when they were imaged. Images were taken using a Zeiss ApoTome with AxioVision software and a 40X objective. Exposure times were kept consistent across animals within each experiment. Statistical comparisons (unpaired t test) were performed using GraphPad Prism 7 software. n = 15 per strain, repeated in duplicate.

## Results

### *ins-4* Regulation in the Hypodermis by SMA-3/Smad

In previous work, we showed that SMA-3/Smad activity in the hypodermis (epidermis) regulates fat accumulation in the *C. elegans* intestine, and that SMA-3/Smad regulation of fat stores depends on the DAF-2 Insulin Receptor (InsR) and is not associated with altered pharyngeal pumping that would modulate feeding rate [1]. We therefore hypothesized that SMA-3/Smad may regulate expression of an insulin-like peptide (ILP) in the hypodermis, which then modulates DAF-2/InsR activity. To identify such an ILP, we considered two transcriptional targets of the DBL-1 pathway previously identified by microarray analysis, *ins-4* and *ins-7* [39]. Of these, *ins-4* is reported to be expressed in hypodermis and neurons, whereas *ins-7* expression has been observed in the intestine and in neurons [40, 41]. To observe INS-4 regulation, we obtained an *ins-4p*::*gfp* reporter and crossed it into a *sma-3* mutant background. In wild-type animals carrying the *ins-4p*::*gfp* reporter, GFP was visible in neurons while only faint GFP expression was observed in hypodermis. However, in *sma-3;ins-4p*::*gfp* animals, GFP expression was starkly increased in the hypodermis, indicating an increase in *ins-4* expression (Fig 1A). We used qRT-PCR to quantify the level of expression of *ins-4* mRNA and found that it is significantly increased in *sma-3* mutant animals compared to controls (mean difference is 7x) (Fig 1B). Thus, three independent methods indicate that DBL-1/BMP signaling normally down-regulates the expression of *ins-4*. Because SMA-3/Smad expression has been detected in the pharynx, intestine, and hypodermis, but not in neurons [42], and because of the striking upregulation of *ins-4p*::*gfp* in the hypodermis in *sma-3* mutants, we conclude that SMA-3/Smad regulates the expression of *ins-4* in the hypodermis.

### Loss of *ins-4* results in a high-fat phenotype

If INS-4 acts as a mediator between Smad signaling in the hypodermis and DAF-2/InsR signaling in the intestine, it may be required for normal lipid accumulation. To date, single mutants for *C. elegans* ILP genes have not displayed mutant phenotypes, due to redundancy between the genes [40, 43]. Nevertheless, we measured lipid accumulation in *ins-4* mutant animals via

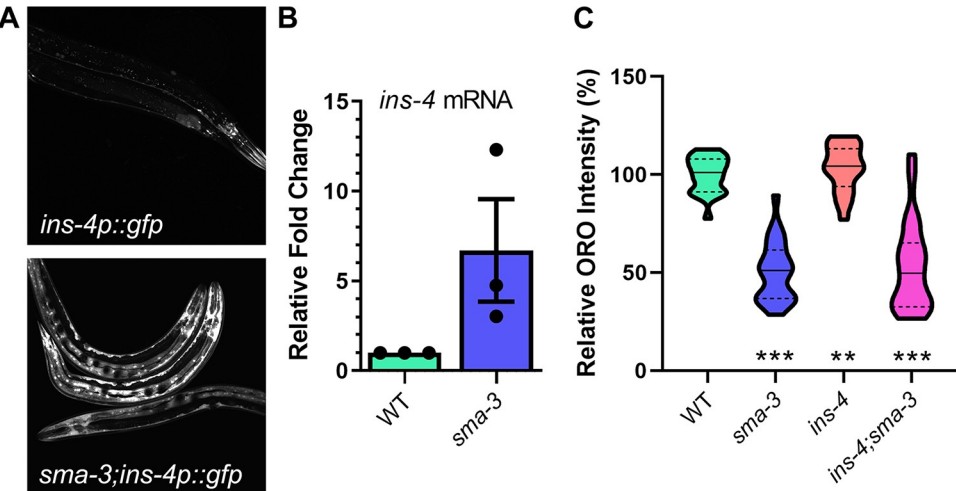

**Fig 1. INS-4 acts downstream of SMA-3 and regulates lipid metabolism.** A) Images of animals expressing *ins-4p*::
GFP. Control animals displayed *ins-4p*::GFP in head neurons, with faint expression in the hypodermis. Upon loss of
*sma-3*, *ins-4p*::GFP expression is significantly increased in the hypodermis. Images taken at 200X, camera settings were
kept constant between all strains. B) Relative quantification of *ins-4* mRNA expression in *sma-3* mutants compared to
control animals via RT-PCR. C) Loss of *ins-4* results in an increase in intestinal lipid stores via Oil Red O staining.
Animals were grown at 20˚C and stained at the L4 larval stage. Quantification was done for equivalent regions of the
intestine just posterior to the pharynx. For all graphs, asterisks across the bottom denote significance compared to
Control, n.s. not significant, * p value < 0.05, ** p value < 0.01, *** p value <0 .001, solid lines denote the median,
dashed lines denote quartiles.

Oil Red O (ORO) staining. We observed an increase of ~20% in the *ins-4* mutants compared
to wild-type (p = 0.0016) (Fig 1C), similar to the increase seen in *daf-2* mutants [1, 19]. We
therefore conclude that INS-4 represses fat accumulation and is required for normal lipid
stores. These results indicate that the loss of *ins-4* is sufficient to alter significantly the level
of lipids in the intestine. As previously published, we saw a significant decrease in lipid accu-
mulation in *sma-3* animals by ~42% (p<0.0001) (Fig 1C). A similar reduction in lipid accumu-
lation was seen in *sma-3* animals at Day 1 and Day 5 of adulthood (S1 Fig). To test the
interaction between *sma-3* and *ins-4*, we generated a *sma-3;ins-4* double mutant and deter-
mined its phenotype. If INS-4 were the only ILP acting downstream of SMA-3, we would
expect the *ins-4* high-fat phenotype to be epistatic to the *sma-3* low-fat phenotype, similarly to
the *daf-2/InsR* high-fat phenotype as previously described [1]. Instead *sma-3;ins-4* double
mutants have a low fat accumulation phenotype compared to wild type (p<0.0001) (Fig 1C),
similar to the low-fat phenotype of *sma-3* single mutants. Therefore, unlike *daf-2/InsR*, *ins-4/
ILP* mutants are not epistatic to *sma-3/Smad* mutants, suggesting that additional ILPs are also
involved in this interaction and can compensate for the loss of *ins-4* in the *sma-3* mutant
background.

## Loss of DBL-1/BMP signaling alters localization of transcription factors downstream of DAF-2/InsR

To elucidate further the mechanisms by which DBL-1/BMP interacts with DAF-2/InsR, we
explored interactions with transcription factors downstream of DAF-2. When DAF-2/InsR is
activated, both DAF-16/FoxO and SKN-1/Nrf are phosphorylated and excluded from the
nucleus; however, upon down-regulation of DAF-2/InsR, both transcription factors are shut-
tled into the nucleus [20, 23]. To observe any alterations to this translocation, we crossed
DAF-16::GFP and SKN-1::GFP transgenes into a *dbl-1* mutant background and treated

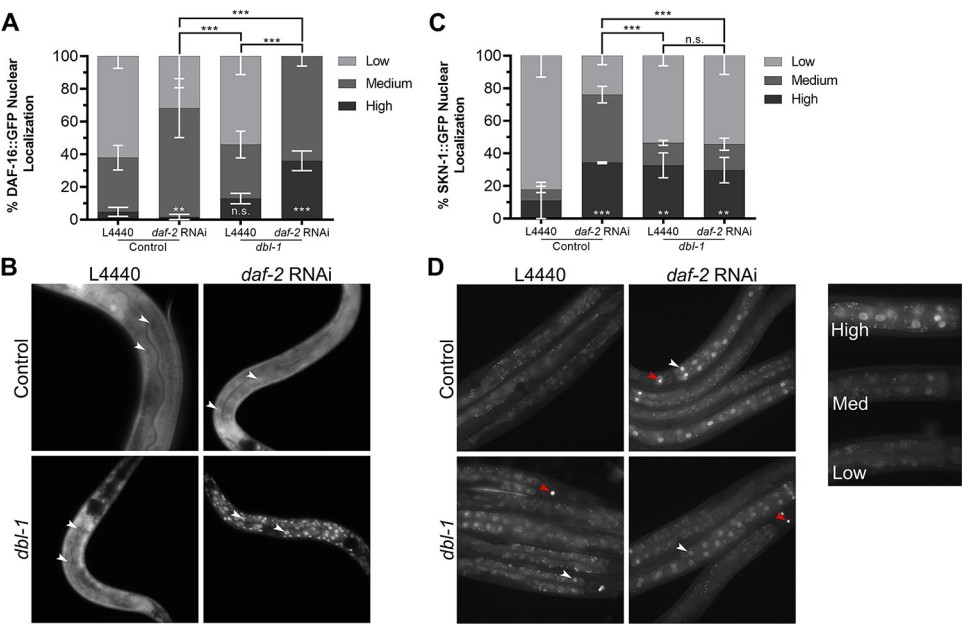

**Fig 2. DAF-16 and SKN-1 localization are altered in a *dbl-1* mutant background.** A) Control animals treated with empty vector exhibit some nuclear localization of DAF-16::GFP, upon treatment with *daf-2* RNAi, DAF-16::GFP nuclear localization increases. In *dbl-1* mutants, DAF-16::GFP nuclear localization is not significantly changed compared to the control under empty vector treatment. Upon treatment with *daf-2* RNAi, a higher level of nuclear localization is observed in *dbl-1* mutants as compared to control animals treated with *daf-2* RNAi. B) Images of animals expressing DAF-16::GFP taken at 400X. White arrows indicate examples of intestinal nuclei. C) Control animals treated with empty vector exhibit very little nuclear localization of SKN-1::GFP. Upon treatment with *daf-2* RNAi, SKN-1::GFP nuclear localization drastically increases. In *dbl-1* mutants, SKN-1::GFP nuclear localization is slightly increased compared to the control under empty vector treatment. *dbl-1* mutants treated with *daf-2* RNAi displayed no change in nuclear localization. D) Images of animals expressing SKN-1::GFP were taken at 200X. White arrows indicate examples of nuclei, red arrows indicate SKN-1::GFP expression in ASI neurons. Nuclear localization was graded as high, medium, or low, with an example of each ranking shown. Asterisks across the bottom represent significance compared to Control, n.s. not significant, $^{**}$ p<0.01, $^{***}$ p<0.001, error bars represent SEM. Camera settings were identical between all strains.

animals with *daf-2* RNAi. Localization of the two transcription factors was observed in the intestinal nuclei due to their size and prominence.

We categorized nuclear localization of the GFP fusion proteins in these backgrounds as low (cytoplasmic only or diffuse in both cytoplasm and nucleus), medium (faint nuclear localization), or high (strong nuclear localization). In a wild-type background, animals treated with the empty vector, L4440, showed DAF-16::GFP fluorescence diffusely or weakly nuclear, with a small percentage showing only cytoplasmic localization (Fig 2A and 2B). As expected, treatment with *daf-2* RNAi caused nuclear accumulation of DAF-16::GFP in all animals (p = 0.0017). In *dbl-1* mutant animals on control RNAi plates, there was no significant difference in the level of nuclear localization compared with the wild-type controls (Fig 2A). Upon reduction of *dbl-1* and *daf-2* together, however, intensity of nuclear DAF-16::GFP became higher than in *daf-2(RNAi)* animals alone (p < 0.0001) (Fig 2A). We conclude that loss of *dbl-1* increases the responsiveness of DAF-16::GFP to the reduction of DAF-2/InsR activity.

In a wild-type background, animals treated with the empty vector, L4440, exhibited mainly diffuse expression of SKN-1::GFP, with very few animals containing nuclear localization. Upon treatment with *daf-2* RNAi, the majority of animals showed a significant increase in nuclear SKN-1::GFP, as expected (p<0.0001). When treated with L4440, *dbl-1* mutants exhibited an increased level of nuclear localization compared to the wild-type control (p = 0.0073),

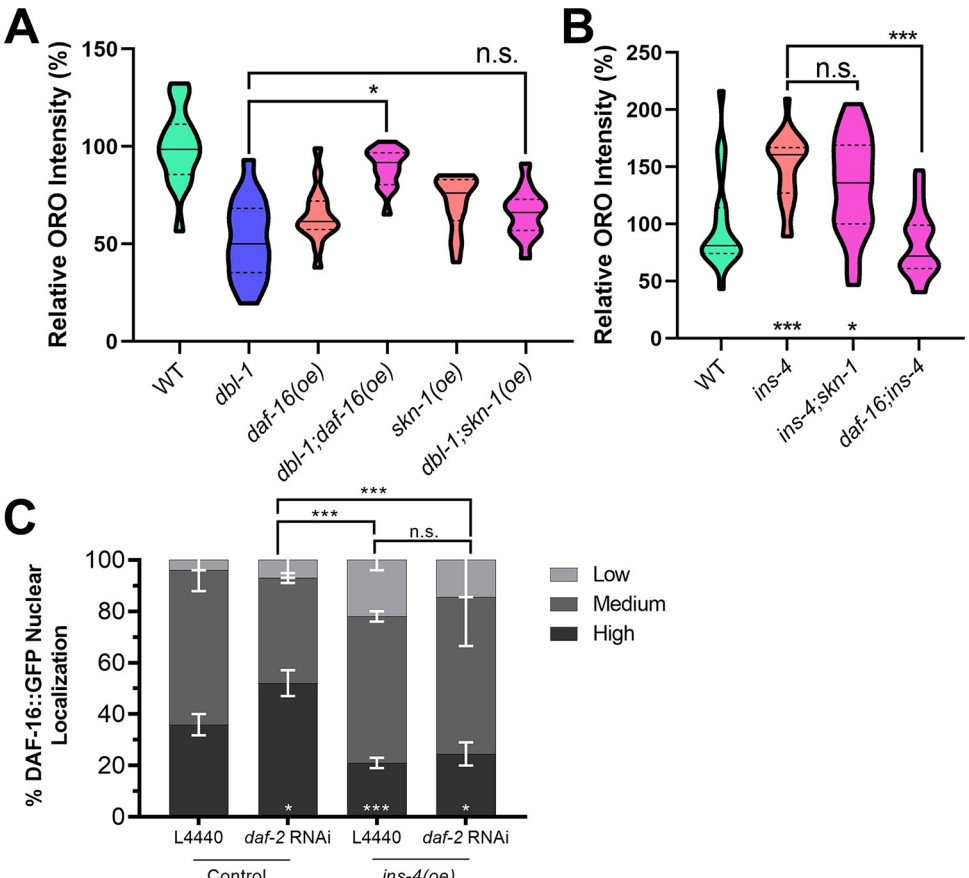

**Fig 3. DAF-16/FoxO but not SKN-1/Nrf mediates BMP-insulin regulation of lipid accumulation.** A) Overexpression of DAF-16::GFP rescues the low-fat phenotype of *dbl-1* mutants while overexpression of SKN-1::GFP does not. B) Loss of *daf-16* blocks the high-fat phenotype of *ins-4* mutant animals. Animals were stained at L4 stage with the neutral lipid dye Oil Red O. Images taken at 400X, camera settings were kept constant between all strains. Quantification was done for equivalent regions of the intestine just posterior to the pharynx. n.s. not significant, * p value < 0.05, *** p value < 0.001, solid lines denote the median, dashed lines denote quartiles. C) DAF-16::GFP nuclear localization in control and *ins-4(oe)* strains with empty vector or *daf-2(RNAi)* treatment.

but less than that of wild type treated with *daf-2* RNAi (p<0.0001). Interestingly, upon treatment with *daf-2* RNAi, localization of SKN-1::GFP in *dbl-1* mutants was not significantly different from those treated with the empty vector (p = 0.8694) (Fig 2C and 2D). This outcome contrasts with the results with DAF-16::GFP. Instead, the responsiveness of SKN-1::GFP to the reduction of DAF-2/InsR was reduced in the *dbl-1* mutant background.

We sought to determine whether one or both of these transcription factors mediate regulation of fat storage downstream of DBL-1/BMP and DAF-2/InsR. The strains expressing tagged functional DAF-16/FoxO or SKN-1/Nrf contain multicopy arrays that may cause an increase in the expression levels of these transcription factors. We measured fat accumulation using ORO staining in L4 stage animals in these strains. In the wild-type background, neither *daf-16*::*gfp* nor *skn-1*::*gfp* have increased fat stores (Fig 3A). Next, we examined *dbl-1* mutant animals. As expected, *dbl-1* mutants have low fat stores. Interestingly, *daf-16*::*gfp* expression, but not *skn-1*::*gfp* expression, is capable of suppressing the low-fat phenotype of *dbl-1* mutants (observed in two out of three trials). This result suggests that DAF-16, but not SKN-1, mediates the effect of DBL-1 on fat accumulation.

## DAF-16/FoxO mediates INS-4/ILP regulation of fat accumulation

To determine whether INS-4/ILP functions through DAF-16/FoxO and/or SKN-1/Nrf to regulate fat accumulation, we determined the lipid accumulation phenotypes of *ins-4* double mutants with *daf-16* and with *skn-1*. As observed previously, loss of *ins-4* alone results in an increased in lipid stores via ORO. In the *daf-16;ins-4* double mutant, the high-fat phenotype of *ins-4* is suppressed by the loss of *daf-16*. However, *ins-4;skn-1* animals exhibit no significant difference from *ins-4* single mutants (Fig 3B). This result provides further support to suggest that INS-4 affects lipid accumulation through the DAF-16/FoxO arm downstream of DAF-2/InsR. We therefore tested the effect of *ins-4* overexpression on DAF-16/FoxO nuclear localization. *ins-4* overexpression reduces DAF-16/FoxO nuclear localization under both control conditions (p = 0.0003) and upon *daf-2/InsR* RNAi (p = 0.0002; Fig 3C). This effect is consistent with the role of INS-4/ILP as an agonist that stimulates DAF-2/InsR activity, increasing DAF-16/FoxO phosphorylation and cytoplasmic retention of DAF-16. The effect on DAF-16/FoxO of increased INS-4/ILP is distinct from that of loss of DBL-1/BMP, suggesting that DBL-1/BMP impinges on DAF-16 nuclear localization through additional mechanisms.

## DBL-1/BMP signaling promotes dauer arrest in a DAF-2/InsR-sensitized background

Because DBL-1/BMP modulates DAF-16/FoxO and SKN-1/Nrf responses to DAF-2/InsR, we might expect it to impinge on other functions of IIS in *C. elegans*. We therefore tested the interactions between these pathways in three functions of IIS: dauer development, aging, and autophagy induction. During development, animals exposed to harsh environmental conditions prior to the third larval stage can activate the stress-adaptive and growth-arrested dauer larval program. IIS is one of the primary regulators of the dauer decision, and *daf-2(e1370)* mutants have a temperature-sensitive dauer-constitutive phenotype. We investigated the degree of dauer formation in animals containing mutations in BMP and IIS pathways at the restrictive temperature, 25°C, and the semi-permissive temperature, 20°C. Normally, the DBL-1/BMP pathway has no effect on dauer, as *dbl-1* single mutants, and other mutants in the pathway, have no observable dauer phenotype at any temperature. As expected, when we placed *sma-3* or *lon-2* mutants at 25°C, we observed no dauer arrest phenotype, as in our wild-type control. When *daf-2* single mutants were placed at the restrictive temperature, we observed dauer arrest in 98.4±0.7% of the progeny, similar to previously published data. This phenotype was also observed in the *daf-2sma-3* and *daf-2;lon-2* double mutants, with 99.5 ±0.6% and 97.6±1.7% dauer formation, respectively.

When animals were grown at the semi-permissive temperature, 20°C, however, DBL-1/BMP signaling had an observable impact on arrested animals in the *daf-2* mutant background. Again, the *sma-3* and *lon-2* mutants and wild type had no dauer induction at 20°C. *daf-2* single mutants exhibited the dauer arrest phenotype in 54.5±6.3% of the progeny at the semi-permissive temperature. The *daf-2sma-3* double mutants, in contrast, exhibited the dauer arrest phenotype in only 23.9±4.6% of the animals, while the *daf-2;lon-2* double mutants exhibited the dauer arrest phenotype in 73.4±7.3% of the animals (Table 1). Two-way ANOVA indicates a significant (p = 0.0006) interaction between the two pathways. This result indicates that DBL-1/BMP signaling has a significant effect on the dauer arrest phenotype in the DAF-2/InsR-sensitive background. DBL-1/BMP signaling appears to modulate dauer formation in a dose-dependent manner, where reduced signaling, as in *daf-2sma-3* double mutants, reduces the occurrence of dauer-like animals, while increased signaling, as in *daf-2;lon-2* mutants, increases the occurrence of dauer-like animals.

**Table 1. DBL-1 Pathway Modulates Dauer Constitutive Phenotype of *daf-2* Mutants at 20°C.** In an otherwise wild-type background, DBL-1 signaling exerts no visible effect on dauer induction at both 20°C and 25°C. However, in a DAF-2-sensitized background, DBL-1 signaling promotes dauer arrest at 20°C. This effect is lost when observed at 25°C, where the loss of DAF-2 signaling is most potent. Values given are percent arrested dauer-like larvae of total progeny ± SEM.

| Strain | 20°C | | 25°C | |
|---|---|---|---|---|
| | % Arrested | N | % Arrested | N |
| N2 | 0 | 239 | 0 | 228 |
| *sma-3(wk30)* | 0 | 243 | 0 | 178 |
| *lon-2(e678)* | 0 | 269 | 0 | 214 |
| *daf-2(e1370)* | 54.5 ± 4.5 | 273 | 98.4 ± 0.5 | 236 |
| *daf-2(e1370)sma-3(wk30)* | 23.9 ± 3.3 | 200 | 99.6 ± 0.4 | 221 |
| *daf-2(e1370);lon-2(e678)* | 73.5 ± 5.2 | 317 | 97.6 ± 1.2 | 199 |

## Loss of SMA-3/Smad modifies DAF-2/InsR longevity phenotype

Because BMP signaling antagonizes the DAF-2/InsR dauer phenotype under sensitized conditions, we wondered whether the same effect might be seen in the lifespan phenotype of *daf-2*. We determined the lifespan phenotypes of N2 (wild type), *sma-3*, *daf-2*, and *daf-2sma-3* animals at 20°C. As expected, loss of *sma-3* had no effect on lifespan, while *daf-2* mutants were long-lived. We compared the lifespans of *daf-2sma-3* double mutants with those of *daf-2* single mutants, and found that loss of *sma-3* causes a reproducibly different lifespan trajectory compared with *daf-2* single mutants, manifesting as a reduction in median lifespan but no effect on maximum lifespan (Fig 4A; p = 0.05 Mantel-Cox, p = 0.01 Gehan-Breslow-Wilcoxon). These results are consistent with DBL-1/BMP signaling impinging on DAF-2/InsR activity.

## DBL-1/BMP signaling enhances autophagic induction in a DAF-2/InsR-sensitized background

Autophagy is a cellular process involved in the recycling of cellular debris and other material, and is vital for the maintained health of an organism [44]. As a known regulator of autophagy, DAF-2/InsR acts to suppress the induction of autophagosome formation, with *daf-2* mutants exhibiting increased levels of autophagy [30, 45–47]. Additionally, autophagy is required for proper dauer development in *daf-2* mutants [30]. DBL-1/BMP signaling has also been shown to inhibit autophagy, as mutants in the BMP pathway elevate clearance of aggregates of SQST-1/p62::GFP autophagy receptor in mutants defective in ribosomal protein RPL-43 [38]. Animals carrying a mutation in *rpl-43* display an accumulation of SQST-1/p62::GFP due to proteotoxic stress caused by defects in ribosomal function that can be suppressed by the induction of autophagic degradation [38]. We reproduced the effect of BMP signaling on SQST-1::GFP accumulation following RNAi inhibition of SMA-3/Smad in *rpl-43* mutants (S2 Fig).

To analyze interactions between BMP and IIS pathways in autophagy regulation, we used GFP::LGG-1, an autophagosome reporter [37], to measure steady state autophagosome levels in double mutants. LGG-1/LC3 is a protein involved in the formation, elongation, and fusion of autophagosomes [30]. Normally localized in a diffuse cytosolic pattern, when autophagic conditions are induced, GFP::LGG-1 becomes localized to pre-autophagosomal structures and autophagosomes. GFP::LGG-1 positive foci were previously counted in hypodermal seam cells. The seam cells form a row of cells that run the length of the hypodermis on each lateral side of the animal, and undergo remodelling in the development of dauer animals. Thus, seam cells were observed due to their size and ease in imaging. *daf-2* mutants are known to show increased levels of autophagy when grown at the semi-permissive and restrictive temperatures,

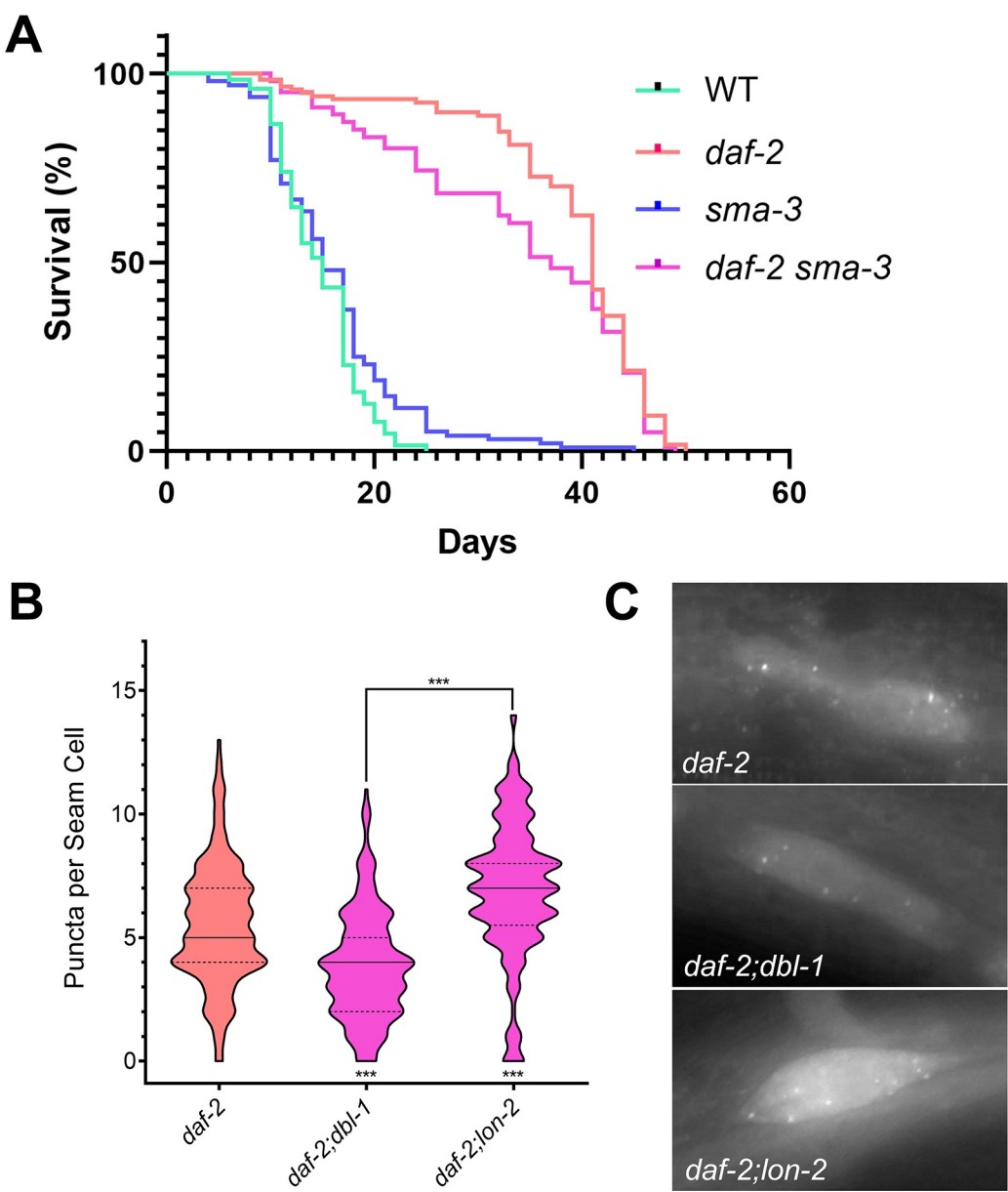

**Fig 4. DBL-1 signaling promotes autophagy in a DAF-2-sensitized background.** A) Loss of *sma-3* induces no noticable effect on lifespan compared to control animals. However, *daf-2;sma-3* animals exhibit a reduced median lifespan compared to *daf-2* single mutants (p = 0.05 Mantel-Cox, p = 0.01 Gehan-Breslow-Wilcoxon). B) Under conditions that induce autophagy such as a reduction in DAF-2 signaling, DBL-1 signaling promotes autophagy, as *daf-2(e1370);dbl-1 (wk70)* mutants show a reduction in GFP puncta, whereas *daf-2(e1370);lon-2(e678)* mutants show an increase in puncta. C) Images of GFP::LGG-1 localization in seam cells of DBL-1 pathway mutants under autophagic induction via *daf-2* loss of function at 630X. All animals were grown at 20˚C and imaged at the L3 larval stage. *** p value < 0.001, solid lines denote the median, dashed lines denote quartiles.

20˚C and 25˚C respectively. To determine if DBL-1/BMP signaling would behave similarly in the regulation of autophagy as it did with dauer formation, we grew the animals at 20˚C. We observed an average of 5.51±2.42 punctae per seam cell in *daf-2* single mutants at 20˚C. In *daf-2;dbl-1* double mutants, the average was decreased to 3.85±2.20 (p<0.0001), while in *daf-2;lon-2* double mutants, the average was increased to 6.90±2.75 (p<0.0001) (Fig 4B and 4C). These results are consistent with the interpretation that DBL-1/BMP activity antagonizes the effect of

the loss of DAF-2/InsR in autophagy induction. Because this assay focuses on steady-state levels of autophagosome production, however, we can't rule out the possibility that loss of *dbl-1* is increasing the speed of clearance of GFP::LGG-1 puncta in *daf-2;dbl-1* double mutants.

The observations presented above, combined with previous work from our lab [1], identifies three modes of interaction between DBL-1/BMP signaling and DAF-2/InsR: independent, epistatic, and antagonistic. DBL-1/BMP and DAF-2/InsR regulate body size independently, as double mutants in the two pathways were observed to have additive effects [1]. Epistatic effects were observed in the regulation of lipids with the high-fat phenotype of *daf-2* mutants completely masking the low-fat phenotype of *sma-3* in *daf-2;sma-3* double mutants. Lastly, the data observed in this study reveals a third relationship between the two pathways; DBL-1/BMP signaling appears to antagonize or partially suppress DAF-2/InsR loss of function phenotypes. These antagonistic effects are only observed in a DAF-2/InsR-compromised background, and not in *daf-2*(+) or when *daf-2(-)* phenotypes are fully penetrant and expressive.

## Discussion

We previously showed that DBL-1/BMP signaling depends on the insulin receptor DAF-2 to regulate lipid stores in *C. elegans* [1]. In this study, we identify components of the IIS pathway that act upstream and downstream of DAF-2 to mediate this interaction. Transcriptome analysis previously identified *ins-4* and *ins-7* as target genes of DBL-1/BMP signaling [39]. Using qRT-PCR and an *ins-4p*::*gfp* reporter, we verified *ins-4* regulation and demonstrated that its expression is highly increased in the hypodermis in a *sma-3* mutant background, indicating that Smad signaling regulates *ins-4* expression in the hypodermis. Additionally, we showed that *ins-4* mutants have increased levels of lipid stores, demonstrating that, in spite of any potential redundancy between ILPs, INS-4 is critically required for regulation of lipid storage. Consistent with our analysis, overexpression of *ins-4* is sufficient to lower fat levels [43]. The other ILP gene we identified as a transcriptional target of DBL-1, *ins-7*, has no effect on fat accumulation when overexpressed [43]. Since *ins-7* is a known transcriptional target of IIS in the intestine [41], its upregulation in *dbl-1* mutants may be a secondary consequence of increased IIS activity.

We also interrogated the effects of downstream IIS transcription factors DAF-16/FoxO and SKN-1/Nrf on DBL-1-mediated fat regulation. Overexpression of DAF-16/FoxO, but not of SKN-1/Nrf, suppresses the effect of DBL-1 on fat accumulation (Fig 3A). Similarly, loss of DAF-16/FoxO, but not of SKN-1/Nrf, suppresses the high-fat phenotype of *ins-4* mutants (Fig 3B). IIS transcription factors DAF-16/FoxO and SKN-1/Nrf are regulated, in part, at the level of subcellular localization. DAF-2/InsR activity promotes cytoplasmic retention of DAF-16/FoxO and SKN-1/Nrf, inhibiting their functions as transcription factors. We found that DBL-1/BMP activity alters the sensitivity of DAF-16/FoxO and SKN-1/Nrf to DAF-2/InsR in opposite ways. For DAF-16/FoxO, we find that there is no change in nuclear accumulation in *dbl-1* mutants compared with control, but that **loss** of *dbl-1/BMP* **increases** the responsiveness of DAF-16/FoxO nuclear accumulation upon RNAi inhibition of *daf-2/InsR* (Fig 3). Conversely, **loss** of *dbl-1/BMP* **decreases** the responsiveness of SKN-1/Nrf to the loss of daf-2/InsR activity. If DBL-1/BMP signaling caused a simple reduction in DAF-2/InsR activity, we would have expected both transcription factors to respond similarly, with increased cytoplasmic localization in a *dbl-1* mutant. DAF-16/FoxO has been shown to act in neurons and the intestine to regulate fat accumulation [48]. We therefore propose a model in which DBL-1 activates Smad signaling in the hypodermis, down-regulating the expression of INS-4, to reduce DAF-2/InsR activity leading to activation of DAF-16/FoxO in the intestine and the maintenance of proper lipid stores (Fig 5).

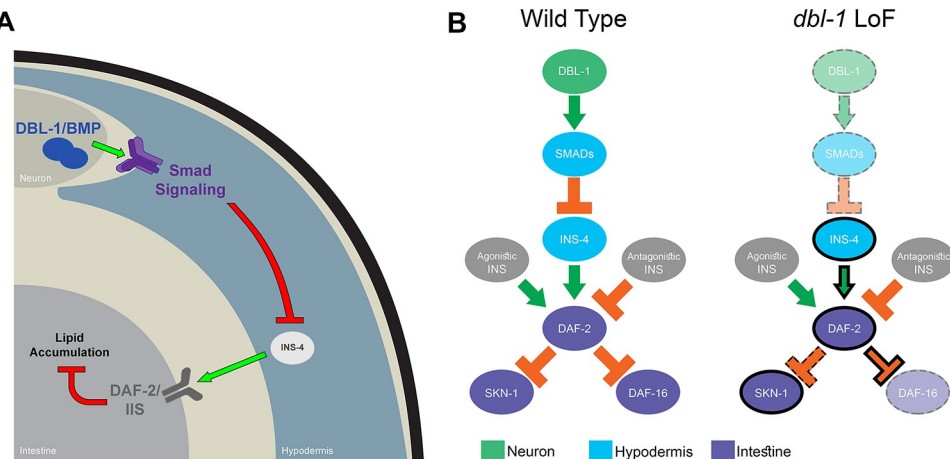

**Fig 5. Model for interaction between DBL-1/BMP signaling and DAF-2/InsR pathways.** A) Smad signaling in the hypodermis regulates the transcription of *ins-4*, which in turn feeds into DAF-2/IIS in the intestine to control lipid metabolsim. B) In wild-type animals, DAF-2/InsR is regulated by multiple agonistic and antagonistic ILPs. DAF-2/ InsR activation leads to phosphorylation and cytoplasmic retention of SKN-1/Nrf and DAF-16/FoxO. In *dbl-1* loss-of-function (LoF) mutants, reduced Smad activity leads to increased INS-4 expression. Also in *dbl-1* mutants, SKN-1/Nrf nuclear localization has reduced sensitivity to DAF-2/InsR activity, while DAF-16/FoxO nuclear localization has increased sensitivity to DAF-2/InsR function.

Altered downstream signaling activation has been demonstrated in other tyrosine kinase receptors, such as Fibroblast Growth Factor (FGF) signaling in mammals [49]. In the FGF signaling family, there exist examples of two ligands with overlapping expression patterns that elicite different responses from the same receptor. Both FGF7 and FGF10 exhibit roles in the branching morphogeneis of the lacrimal and submandibular glands in mice using *ex vivo* organ cultures. These two ligands both bind to FGFR2b (fibroblast growth factor receptor 2b), however, FGF7 induces a proliferative response while FGF10 induces a migratory response, switching morphogenesis between branching and elongation respectively [50, 51]. The mechanism thought to underly this switch is through the differential phosphorylation of a specific tyrosine residue. Binding of FGF10 to FGFR2b results in the phosphorylation of residue Y734, which recruits a complex involved in endocytic recycling, while binding of FGF7 does not [52]. The exact mechanism by which the two ligands result in differential phosphorylation is unknown, however structural evidence suggests that binding of the ligands may induce two different confirmations of the intracellular domains [53]. While this mechanism has not been studied in the context of DAF-2/InsR in *C. elegans*, it does not preclude the possibility that this mechanism may be shared between the two tyrosine kinase receptor families. Further studies on the levels of phosphorylation by DAF-2 in the intact BMP or inactive BMP background are required.

Based on this work and other published studies, we infer the existence of at least three modes of interaction between DBL-1/BMP signaling and DAF-2/InsR: independent, epistatic, and antagonistic. BMP and IIS pathways act independently in body size regulation and reproductive aging [1, 54, 55]. In lipid homeostasis, IIS acts downstream of BMP signaling resulting in an epistatic effect (this work). Finally, DBL-1/BMP signaling pathway exhibits an antagonistic effect on dauer formation, aging, and autophagy induction in a DAF-2/InsR mutant background under sensitized conditions. In addition to the interactions already discussed, Qi et al. identified a cooperative interaction between DBL-1/BMP signaling and DAF-16/FoxO that does not occur at the level of subcellular localization [55]. They find that DAF-16/FoxO functions nonautonomously in the hypodermis to induce germline tumor formation, and that this

function depends on DBL-1/BMP signaling. This phenotype is independent of the nuclear localization of DAF-16/FoxO; rather the Smads SMA-2 and SMA-3 were shown to bind DAF-16 directly and presumably promote its transcriptional activity [55]. In addition, DBL-1/BMP interacts with IIS in *C. elegans* L1 arrest. In *C. elegans*, embryos that hatch without food enter an L1 arrest in which cell divisions are blocked. The arrested somatic development of different cell lineages has been shown to be dependent on DBL-1/BMP interaction with IIS. In mesodermal lineages, the cell division arrest is defective in *daf-16* mutants, and DBL-1/BMP activity is required for the L1 arrest defective phenotype of *daf-16* [56]. In neuronal lineages, DBL-1/BMP acts upstream to promote IIS activity in a DAF-16-independent role, through reduction in the expression of *ins-3* and *ins-4* in neurons [57]. In vertebrates, tissue culture studies have shown that BMP2 and BMP6 regulate insulin sensitivity in adipose cells [8]. In mice, BMP4 regulates insulin sensitivity of adipose tissue [58]. Furthermore, BMP7 regulates expression of insulin signaling components in C3H10T1/2 cells, leading to fat browning [7]. Taken together, these analyses exemplify the interconnected, polymodal nature of BMP and IIS pathway interactions in the regulation of organismal function. Using *C. elegans* we have now elucidated new mechanisms of interaction between these two conserved signaling pathways at the level of ILP and transcription factor regulation in the context of the intact organism.

## Supporting information

**S1 Fig. SMA-3/Smad regulates lipid accumulation through adulthood.** *sma-3* animals exhibit reduced lipid accumulation during adulthood via ORO staining. This reduction is independent of reproduction, as treatment with FUDR prevents the development of eggs. These results suggest that *sma-3* continues to affect lipid accumulation through development and into adulthood. Asterisks across the bottom denote significance compared to Control, n.s. not significant, * p value < 0.05, *** p value <0 .001, solid lines denote the median, dashed lines denote quartiles.
(TIF)

**S2 Fig. Inactivation of *sma-3/Smad* suppresses the accumulation of SQST-1::GFP aggregates in *rpl-43* mutants.** A) *rpl-43(bp399)* mutants show accumulation of SQST-1::GFP puncta due to impaired ribosomal activity [38]. Genetic backgrounds that increase autophagy allow the clearance of those aggregates. We analyzed *rpl-43* mutants at the L4 stage following RNAi depletion of *sma-3/Smad* compared with empty vector control (L4440), and found a significant clearance of SQST-1::GFP puncta (p < 0.001) as expected [38]. Two trials of n = 15 animals yielded similar results, and the combined data are shown. B) Images of *sqst-1*::*gfp* animals treated with either an empty vector control (L4440) or *sma-3* RNAi taken at 400X.
(TIF)

## Acknowledgments

We thank Michael Meade and Shoshana Reich for the construction of CS663 and CS633, respectively. We thank Nick Palmisano for input on autophagy assays. Some strains were provided by the CGC, which is funded by NIH Office of Research Infrastructure Programs (P40OD010440). This work was carried out in partial fulfillment of the requirements for the Ph.D. degree from the Graduate Center of City University of New York (JFC, EJC, KKY, and UM).

## Author Contributions

**Conceptualization:** James F. Clark, Alicia Meléndez, Cathy Savage-Dunn.

**Funding acquisition:** Alicia Meléndez, Cathy Savage-Dunn.

**Investigation:** James F. Clark, Emma J. Ciccarelli, Peter Kayastha, Gehan Ranepura, Katerina K. Yamamoto, Muhammad S. Hasan, Uday Madaan, Cathy Savage-Dunn.

**Project administration:** Cathy Savage-Dunn.

**Supervision:** Alicia Meléndez, Cathy Savage-Dunn.

**Writing – original draft:** James F. Clark, Cathy Savage-Dunn.

**Writing – review & editing:** James F. Clark, Emma J. Ciccarelli, Peter Kayastha, Gehan Ranepura, Katerina K. Yamamoto, Muhammad S. Hasan, Uday Madaan, Alicia Meléndez, Cathy Savage-Dunn.

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
