## [Decision Letter · Decision Letter 0]

10 Sep 2021

Dear Dr Savage-Dunn,

Thank you very much for submitting your Research Article entitled 'BMP Pathway Regulation of Insulin Signaling Components Promotes Lipid Storage in Caenorhabditis elegans' to PLOS Genetics.

The manuscript was fully evaluated at the editorial level and by independent peer reviewers. The reviewers appreciated the attention to an important topic but identified some concerns that we ask you address in a revised manuscript

We therefore ask you to modify the manuscript according to the review recommendations. Your revisions should address the specific points made by each reviewer.

[LINK]

Yours sincerely,

Anne Brunet

Guest Editor

PLOS Genetics

Gregory Barsh

Editor-in-Chief

PLOS Genetics

Reviewer's Responses to Questions

**Comments to the Authors:**

Reviewer #1: Comments to PGENETICS-D-21-01120

Clark and colleagues delineated a genetic interaction and pathway interactions of BMP signaling to Insulin/IGF-1 signaling regulating fat accumulation in C. elegans.

I have read the reviewer responses and the manuscript. To me, the previous reviewers’ requests were adequately addressed. I agree with the previous reviewer suggestion to focus on the most novel findings to gain more mechanistically insights. I also agree with the choice of the authors’ to focus on ins-4 and the interaction with BMP signaling.

Based on their work, if I understand the signaling pathway correctly;

the proposed model is that DBL-1 is released from the neurons to activate SMAD signaling in the hypodermis via sma-3, which inhibits the gene expression of ins-4 in the hypodermis. Insulin-like peptide (ins-4) is then released from the hypodermis to act as an agonist to increase daf-2/Insulin/IGF-1 receptor signaling in the intestine to retain FOXO/DAF-16 in the cytoplasm, and thus, inactive. This somehow leads to fat accumulation.

This genetic model I think is the essence of the story and provides a coherent message for the revised manuscript (although I have not seen the previous version).

I only have two concerns that can be addressed by text changes.

1) Clarification of the genetic model for fat accumulation. Figure 1: Loss of sma-3 increases ins-4 expression levels, suggesting that sma-3 is upstream of ins-4. In the analysis for fat accumulation, loss of sma-3 leads to lower fat levels vs loss of ins-4 mildly increases fat levels, hence opposite phenotypes making it a good system to determine epistasis. One would expect if fat reduction of loss of sma-3 would act through ins-4 (as suggested in the model), then the double mutant of sma-3;ins-4 should phenotypically look like ins-4 (mild increase in fat) but shows the opposite (ie, lower fat) phenocopying sma-3. This suggests that the fat reduction of loss of sma-3 is independent of ins-4. A clarification of this finding and how it fits the model would be great.

2) When I was reading this manuscript, I kept asking myself, how does loss of sma-3 lead to such a strong loss of fat levels. It is a pretty striking phenotype with 50-75% fat loss compared to WT when looking at fig 1 and fig S1. Could the higher or reduced fat accumulation be simply due to changes in feeding rates? Recently, daf-2/daf-16 signaling has been shown to regulated feeding rates (PMID: 30905669). Could the observed changes in fat levels and autophagy be simply due to different feeding rates in dbl-1/sma-3 and ins-4? Would smaller body size not also correlate with lower feeding rates? I understand coming into a revised manuscript as a new reviewer, it is not fair to ask for addition experiments (besides control experiments). Thus, I do not ask for the experiments but rather a discussion as possibility to think about and for perhaps future directions. May be the authors already know good arguments of why feeding rates is not a likely explanation of the observed phenotypes.

Reviewer #2: In this manuscript, Clark et al., identify an insulin-like peptide, ins-4, as a mediator of BMP and daf-2/IIS pathway interactions to regulate lipid metabolism. The data presented are intriguing and will be of interest to the C. elegans community. However, a couple of key experiments are missing that would strengthen the authors’ most novel claim that BMP signaling regulates ins-4 in the hypodermis to regulate fat metabolism.

The authors argue that sma-3 in the hypodermis regulates expression of ins-4, and state that ins-4 is “highly and specifically increased in the hypodermis in the sma-3 mutant. However, the only piece of evidence that ins-4 levels are increased in the hypodermis is based on fluorescent reporter images. It is not clear whether ins-4::GFP fluorescence is also increased in neurons based on the images provided. While I appreciate that they confirmed that ins-4 mRNA is increased in sma-3 mutants, this is done from whole animal lysate, so it is not appropriate to make any claims of tissue specificity. Because this cell non-autonomous communication contributes to the novelty of the manuscript, I think the authors should perform a more direct experiment to determine if INS-4 is originating from hypodermis to regulate lipid metabolism. For example, does tissue specific overexpression of sma-3 in hypodermis (strain published in the author’s previous paper) in the sma-3 mutant background reduce ins-4:GFP in the hypodermis?

In Figure 3C, it is not clear if the ins-4 overexpression is hypoderm-specific. This manuscript would be significantly strengthened if the authors provide data that ins-4 overexpression in the hypodermis induces daf-16 nuclear localization in the intestine and affects lipid metabolism. Ins-4 loss-of-function experiments via ins-4 hypoderm-specific RNAi, for example, would also be informative.

**Have all data underlying the figures and results presented in the manuscript been provided?**

Reviewer #1: Yes

Reviewer #2: Yes

PLOS authors have the option to publish the peer review history of their article (what does this mean?). If published, this will include your full peer review and any attached files.

Reviewer #1: **Yes: **Collin Y. Ewald

Reviewer #2: No

---

## [Editor Report · Decision Letter 1]

27 Sep 2021

Dear Dr Savage-Dunn,

We are pleased to inform you that your manuscript entitled "BMP Pathway Regulation of Insulin Signaling Components Promotes Lipid Storage in Caenorhabditis elegans" has been editorially accepted for publication in PLOS Genetics. Congratulations!

Yours sincerely,

Anne Brunet

Guest Editor

PLOS Genetics

Gregory Barsh

Editor-in-Chief

PLOS Genetics

Comments from the reviewers (if applicable):

**Data Deposition**

http://datadryad.org/submit?journalID=pgenetics&manu=PGENETICS-D-21-01120R1

**Press Queries**

---

## [Editor Report · Acceptance letter]

8 Oct 2021

PGENETICS-D-21-01120R1 

BMP Pathway Regulation of Insulin Signaling Components Promotes Lipid Storage in *Caenorhabditis elegans*  

Dear Dr Savage-Dunn, 

We are pleased to inform you that your manuscript entitled "BMP Pathway Regulation of Insulin Signaling Components Promotes Lipid Storage in *Caenorhabditis elegans* " has been formally accepted for publication in PLOS Genetics! Your manuscript is now with our production department and you will be notified of the publication date in due course.

With kind regards,

Olena Szabo

PLOS Genetics

On behalf of:
